# Genomic Diversity and Recombination Analysis of the Spike Protein Gene from Selected Human Coronaviruses

**DOI:** 10.3390/biology13040282

**Published:** 2024-04-22

**Authors:** Sayed Sartaj Sohrab, Fatima Alsaqaf, Ahmed Mohamed Hassan, Ahmed Majdi Tolah, Leena Hussein Bajrai, Esam Ibraheem Azhar

**Affiliations:** 1Special Infectious Agents Unit, King Fahd Medical Research Center, King Abdulaziz University, P.O. Box 80216, Jeddah 21589, Saudi Arabia; fha424@gmail.com (F.A.); hmsahmed@kau.edu.sa (A.M.H.); atoulah@kau.edu.sa (A.M.T.); lbajrai@kau.edu.sa (L.H.B.); 2Department of Medical Laboratory Sciences, Faculty of Applied Medical Sciences, King Abdulaziz University, P.O. Box 80216, Jeddah 21589, Saudi Arabia; 3Department of Medical Laboratory Sciences, Faculty of Applied Medical Science, King Abdulaziz University, P.O. Box 21911, Rabigh 344, Saudi Arabia; 4Biochemistry Department, Faculty of Sciences, King Abdulaziz University, P.O. Box 80216, Jeddah 21589, Saudi Arabia

**Keywords:** genetic diversity, phylogeny, recombination, spike protein, MERS-CoV, SARS-CoV-2, HCoVs

## Abstract

**Simple Summary:**

Coronaviruses are serious pathogens for both humans and animals. The name corona was designated because of the crown-like spikes on their surface. Currently, seven coronaviruses have been identified, such as 229E, NL63, OC43, HKU1, MERS-CoV, SARS-CoV, and SARS-CoV-2. Sometimes, animal coronaviruses infect humans and evolve due to genetic mutations, interspecies transmission, host adaptations, and favorable conditions. The main objective of this study was to analyze the genetic diversity and predict the emergence of new variants with novel properties. It has been reported that the spike protein gene plays an important role in host cell attachment and entry into host cells. The S gene has the highest mutation/deletion and is the most utilized target for vaccine/antiviral development. In this work, we discussed the genetic diversity, phylogenetic relationship, and recombination patterns of selected HCoVs with an emphasis on the newly emerged SARS-CoV-2 and MERS-CoV. The findings of this study showed that MERS-CoV and SARS-CoV-2 have significant sequence identities with the selected HCoVs. The phylogenetic and recombination results concluded that new variants may emerge in the future with novel properties that infect both humans and animals. This information will be helpful for global society to design and develop an effective vaccine and disease management strategy.

**Abstract:**

Human coronaviruses (HCoVs) are seriously associated with respiratory diseases in humans and animals. The first human pathogenic SARS-CoV emerged in 2002–2003. The second was MERS-CoV, reported from Jeddah, the Kingdom of Saudi Arabia, in 2012, and the third one was SARS-CoV-2, identified from Wuhan City, China, in late December 2019. The HCoV-Spike (S) gene has the highest mutation/insertion/deletion rate and has been the most utilized target for vaccine/antiviral development. In this manuscript, we discuss the genetic diversity, phylogenetic relationships, and recombination patterns of selected HCoVs with emphasis on the S protein gene of MERS-CoV and SARS-CoV-2 to elucidate the possible emergence of new variants/strains of coronavirus in the near future. The findings showed that MERS-CoV and SARS-CoV-2 have significant sequence identity with the selected HCoVs. The phylogenetic tree analysis formed a separate cluster for each HCoV. The recombination pattern analysis showed that the HCoV-NL63-Japan was a probable recombinant. The HCoV-NL63-USA was identified as a major parent while the HCoV-NL63-Netherland was identified as a minor parent. The recombination breakpoints start in the viral genome at the 142 nucleotide position and end at the 1082 nucleotide position with a 99% CI and Bonferroni-corrected *p*-value of 0.05. The findings of this study provide insightful information about HCoV-S gene diversity, recombination, and evolutionary patterns. Based on these data, it can be concluded that the possible emergence of new strains/variants of HCoV is imminent.

## 1. Introduction

Coronaviruses (CoVs) fall under the *Coronviridae* family [1]. This family consists of ss +ve sense RNA viruses, which are separated based on their phylogeny into four genera: alpha-CoV, beta-CoV, gamma-CoV, and delta-CoV [2]. Generally, alpha- and beta-CoVs mainly include CoVs of mammalian origin, while the gamma- and delta-CoVs commonly include CoVs of avian origin [3,4,5]. Structurally, all CoVs have a similar organization of their genomes, being approximately 26–32 kb in size with varied G+C contents of 32% to 43%. The major part of the viral genome contains two open reading frames (ORFs) encoding 16 non-structural proteins. The remaining portion contains the spike (S), membrane (M), envelope (E), and nucleocapsid (N) proteins, which are encoded by other ORFs, as seen in Figure 1 [2,6]. Based on the current reports, human coronaviruses (HCoVs) are well known to transmit easily to other species. Seven HCoVs have emerged so far, causing serious illnesses ranging from mild self-limiting symptoms like the common cold to life-threatening conditions like severe respiratory syndromes [6]. For years, HCoVs such as HKU-1, NL63, 229E, and OC43 were not considered serious human pathogens as they only caused mild illnesses. The first identification of HCoV-HKU1 was completed in 2005 from a patient with pneumonia symptoms in China [7]. HCoV-NL63 was detected for the first time in 2004 in a Dutch child [8]. HCoV-229E was identified in 1966 and finally isolated in 1967 [9]. In 1960, the isolation of HCoV-OC43 was completed from human tracheal explants. SARS-CoV-1 was identified from China in 2002 and designated as the first highly pathogenic HCoV [10,11]. The civet cat was identified as a primary host and due to human-to-human transmission, this virus spread to 26 countries, resulting in a global epidemic that resulted in 8098 cases and 774 deaths by July 2003. This virus disappeared within eighteen months, and no more cases were reported after January 2004 [6,11]. The second pathogenic MERS-CoV was identified from a 60-year-old patient in Jeddah, the Kingdom of Saudi Arabia, in 2012, and to date this virus has been reported in 27 countries [12]. MERS-CoV causes zoonotic respiratory disease and is currently known as a serious pathogen for both humans and camels [13]. Bats and dromedary camels have been identified as primary source for human infection [5,14,15,16,17]. MERS-CoV caused an outbreak in the Arabian Peninsula, African countries, and South Korea, and resulted in more than 2605 cases with 937 deaths [18,19,20,21,22]. The genomic alterations and favorable conditions in a specific location may lead to the re-emergence of pathogenic HCoVs and human infections [4,6]. In late December 2019, the third human pathogenic SARS-CoV-2 emerged because of favorable climatic conditions in Wuhan city, China, that resulted in a global pandemic [23]. As of today, SARS-CoV-2 has spread into 231 countries, with 704,753,890 confirmed cases and 7,010,681 deaths, as well as 675,619,811 recoveries (https://www.worldometers.info/coronavirus/—last accessed on 17 April 2024, 01:00 GMT). All HCoVs are zoonotic viruses that circulate among different animal species before infecting humans. Several pieces of evidence support the theory that most of the HCoVs originated in bats, where they became well adapted [6]. Interspecies transmission of HCoVs and animal coronaviruses continues to increase their genetic diversity and evolutionary rate, resulting in the emergence of novel coronaviruses with diverse characteristics and extended hosts [24,25]. The family *Coronaviridae* undergoes both homologous and non-homologous recombination, genetic mutation, insertion, and deletion. Among HCoVs, the pattern of recombination was observed for the first time in SARS-CoV-1 in 2004. Additionally in 2006 the recombination was identified in HCoV-HKU1 and HCoV-NL63, followed by the recombination reports in 2011 and 2014 for HCoV-OC43 and MERS-CoV, as well as recently in SARS-CoV-2 in 2020 [26]. The HCoVs-Spike (S) gene has the highest mutation rate site, insertion, and deletion, and has been the most used target for vaccine/antiviral development. The S gene has been identified as being key for host cell attachment and facilitating host cell entry [27]. In MERS-CoV, the S gene attaches DPP4 and CD26 for host cell entry through the receptor-binding domain (RBD), which mediates the interaction, while ACE2 has been identified as the S gene receptor for SARS-CoV-1 and SARS-CoV-2 [28,29,30,31]. In a recent recombination study, co-infections with different MERS-CoV lineages have been reported [22]. Based on recently analyzed samples from Uganda, it was observed that there were many amino acid substitutions in the RBD and recombination in the S1 sub-unit of the S protein gene, which may have facilitated the emergence of MERS-CoV and caused human disease [29,32]. Several significant variations have been identified in the non-structural and structural genes of MERS-CoV among humans and camels, which have significantly impacted virus transmission, disease spread, and the evolution of the virus in various geographical locations, resulting in the emergence of recombinant viruses [6,22,33,34,35,36].

Currently, many reports have been published about recombination in SARS-CoV-2, which has resulted in the emergence of variants of concern (VOCs) and variants of interest (VOIs) (https://www.who.int/activities/tracking-SARS-CoV-2-variants, accessed on 17 April 2024, https://www.ecdc.europa.eu/en/COVID-19/variants-concern, accessed on 17 April 2024) [2,37,38,39,40,41]. In March 2020, Li et al. reported that the whole RBD of the S gene was introduced through recombination with coronaviruses from pangolins [26], and this was further validated by Zhu et al. in December 2020 [42]. However, based on a recent study using sliding-window bootstrapping, SARS-CoV-2 was defined as a mosaic genome with three bat SCoV2rCs reported from Yunnan, China [4,41,43]. The S gene of SARS-CoV-2 has also been found to have many variants that can affect the virus transmissibility, infectivity, and vaccination efficacy. These variants were classified as variants of interest (VOIs), such as the Lambda and Mu variants, while others were classified as variants of concern (VOCs), such as the Alpha, Beta, Gamma, Delta, and Omicron variants [30,44]. It is essential to consider that the classifications of variants can be changed according to recent updates from global studies [40,44]. Therefore, identifying the genetic diversity of the HCoVs-S gene is essential to understand how evolution can affect the viral pathogenesis and transmission of HCoVs with altered properties to new hosts. Based on the recent status, we performed this work to elucidate the genomic diversity and recombination pattern of the selected HCoVs-S gene. The S gene plays an important role in virus–host cell attachment and entry into infected cells. The S gene has been widely used for vaccine/antiviral development against HCoVs. The main goals of this study were to perform genetic diversity, phylogeny, and recombination pattern analyses of SARS-CoV-2 and MERS-CoV along with other HCoVs. Additionally, we extended our objectives to identify the possible emergence of new variants/strains of HCoV in the near future. This detailed information about the genetic diversity, phylogeny, and recombination pattern of the selected HCoVs-S gene could be extensively used by the scientific community as well as disease control authorities to protect the global human population by designing effective vaccines and antivirals for long-term broad-spectrum protection from coronavirus infections.

## 2. Materials and Methods

### 2.1. Retrieval of Viral Genome Sequences

The selected HCoVs-S protein gene sequences from different hosts and locations were retrieved from GenBank, NCBI-PubMed. We included the highest number of S protein gene sequences from SARS-CoV-1, SARS-CoV-2, and MERS-CoV, followed by other HCoVs. A total of 19 sequences of SARS-CoV-2; 19 sequences of SARS-CoV-1; 26 sequences of MERS-CoV; 16 sequences of HCoV-NL63, HCoV-229E, and HCoV-OC43, each; and 8 sequences of HCoV-HKU-1 from different hosts and geographic locations were collected. For the genetic analysis of MERS-CoV with SARS-CoV-2, we selected mostly from the Arabian Peninsula, while for the analysis of SARS-CoV-2 with other HCoVs, we selected and divided the sequences based on their identification and global distribution from multiple hosts and locations. As it has been reported that the MERS-CoV is more prevalent in the Arabian Peninsula than other locations, we focused on the collection and division of these sequences from the Arabian Peninsula region. The selection of viral sequences was made via filtration based on their geographical distributions and spread, as well as their frequency of prevalence globally. Our objective was to collect and analyze the S protein gene sequences of the most prevalent viruses and their number of laboratory-confirmed cases, as well as deaths reported globally. We used SARS-CoV-2 (Accession# MW837148) and MERS-CoV (Accession# NC_019843) as reference sequences to perform all of the analyses because MERS-CoV and SARS-CoV-2 have shown high sequence identity together.

### 2.2. Genome Analyses of HCoVs

The S protein gene sequences of the selected HCoVs (nucleotide—[NT] and amino acid—[AA]) were aligned by using the CLUSTALW and BioEdit (v.7.2.5) online software tools. The sequence similarity and identity matrices were determined based on the MERS-CoV (Accession# NC_019843) and SARS-CoV-2 (Accession# MW837148) genomes as reference sequences with other HCoVs collected from various parts of the world. To identify the phylogenetic relationships of MERS-CoV and SARS-CoV-2 sequences with other HCoV genomes, the MEGA11 software program was used and a phylogeny dendrogram was generated [45]. Initially, the phylogenetic analysis was performed by using the genome sequences of all HCoVs together. Then, we performed another phylogenetic analysis by using MERS-CoV and HCoVs without the SARS-CoV-2 genome, as well as the nucleotide sequences of SARS-CoV-2 sequences with all selected HCoVs without MERS-CoV. We also performed a phylogenetic analysis using only MESRS-CoV with SARS-CoV-2 and SARS-CoV-1 genome sequences.

### 2.3. Recombination Pattern Analyses among HCoVs

The selected S protein gene sequences of HCoVs were used to analyze the recombination pattern and elucidate possible recombinants among the minor and major parents by using the recombination detection program (RDP v. 5 program) [46]. The SARS-CoV-2 (Accession# MW837148) S protein gene sequence was used as a reference sequence. The generated FASTA files were exported to the RDP v. 5 program for analysis, and the recombinants with recombination breakpoints and hot spots, including their start and end points in the viral genome, were identified using the software and putative recombination events were identified in the S protein gene sequences of SARS-CoV-2 (MW837148). The putative recombination events were identified with a cut-off *p*-value (≤0.05).

## 3. Results

### 3.1. Genome Analyses of HCoVs

The SARS-CoV-2-S protein gene (Accession# MW837148) sequence was used as a reference sequence to analyze the sequence identity based on nucleotide (NT) and amino acid (AA) sequences with selected HCoV sequences. The highest sequence identities (99.9%—NT and 99.8%—AA) were identified with multiple SARS-CoV-2 isolates, while the lowest identities (32.3%—NT and 19.8%—AA) were identified with an isolate of HCoV-229E-USA (Accession# KY369914). The sequences (NT/AA) from SARS-CoV-2 showed higher sequence identities when compared to MERS-CoV and others such as HCoV-229E, HCoV-OC43, HCoV-NL63, and HCoV-HKU1 collected from various locations during different periods (Table 1).

Additionally, we also analyzed the percent sequence identity matrix based on the NT/AA sequence of the MERS-CoV-S protein gene (Accession# NC_019843) as a reference sequence with only the selected HCoVs. The percent sequence identity matrix ranged from 99.7 to 99.3% for the NTs and from 99.9 to 99.4% for the AAs with the selected MERS-CoV, while the NT sequences of the remaining HCoVs, along with HCoV-OC43, HCoV-NL63, HCoV-HKU1, and HCoV-229E, ranged from 46.2 to 34.6%, and the AA sequences ranged from 28.9 to 18.1% with MERS-CoV (Table 2).

An analysis was also performed by using the SARS-CoV-2-S protein gene sequence (NT/AA) as a reference sequence with selected HCoVs. The highest identity (99.9%—NT) was observed, while the lowest was 98.5%, and the AA identity varied from 99.9% to 97% with SARS-CoV-2 from various regions. SARS-CoV-1 showed the highest similarity (73.0% NT), the lowest was 72.9%, and the AA identity ranged from 75.8 to 75.4% with SARS-CoV-2. The other HCoVs, such as HCoV-OC43, HCoV-NL63, HCoV-HKU1, and HCoV-229E, showed a variable range of identity for both the NTs and AAs with SARS-CoV-2 (Table 3).

Another analysis was performed by only using the NT/AA sequence of the MERS-CoV-S protein gene as a reference sequence along with the selected SARS-CoV-1 and SARS-CoV-2-S protein genes. The highest NT identity ranged from 99.7 to 99.2%, and the AA identity ranged from 99.9 to 99.7% for MERS-CoV. The NT sequence identity for SARS-CoV-2 ranged from 44.2 to 44.0%, while the AA identity was 26.7–26.6%. SARS-CoV-1 showed a variable range of sequence identity, which ranged from 44.9 to 44.8% for the NTs and from 26.6 to 26.4% for the MERS-CoV isolates (Table 4).

### 3.2. Phylogenetic Analyses

Phylogenetic tree analyses were performed using the nucleotide (NT) sequences of the S protein gene sequences with selected HCoVs. The sequence of SARS-CoV-2 (MW837148) was used as a reference sequence to perform a phylogenetic tree analysis with other viral sequences. The phylogeny was generated by using NT sequences separated into different clusters. All of the HCoVs clustered separately and formed closed clusters with their similar isolates. SARS-CoV-2 (MW837148) only formed a closed cluster with the SARS-CoV-2 isolates selected from different locations (Figure 2).

Additionally, a phylogenetic analysis of the MERS-CoV-S protein gene with other HCoVs was performed by using MERS-CoV (KF958702) as the reference sequence. The results showed that multiple clusters were formed with the selected HCoVs. All MERS-CoV samples formed a closed cluster with the selected MERS-CoV isolates. Interestingly, similar clustering was observed with the other HCoVs and their respective virus isolates (Figure 3).

In another analysis, a similar phylogenetic relationship analysis was performed by excluding the MERS-CoV isolates. SARS-CoV-2 (MW837148) was used as a reference and analyzed with the SARS-CoVs and the selected HCoVs. SARS-CoV-2 (MW837148) clustered with only the selected SARS-CoV-2 isolates; interestingly, none of the SARS-CoV-1 isolates clustered with any SARS-CoV-2. SARS-CoV-1 formed a separate cluster, and similarly, all of the selected HCoVs clustered separately (Figure 4).

Finally, one more phylogenetic tree analysis was performed by using the MERS-CoV (KF958702) S protein gene sequence as a reference sequence with only SARS-CoV-1 and SARS-CoV-2. It was observed that all of the MERS-CoV samples formed a separate cluster. Interestingly, both SARS-CoV-1 and SARS-CoV-2 clustered separately in this phylogenetic tree analysis (Figure 5).

### 3.3. Recombination Analyses

The genome of the SARS-CoV-2-S protein gene sequence (MW837148) was used as a reference sequence to elucidate the pattern of recombination with selected HCoVs, including MERS-CoV. Putatively, two recombination breakpoints were generated by using the RDP v. 5 program (Figure 6a). Respectively, for all of the sequences analyzed, HCoV-NL63-USA (Accession# JQ771059) was identified as a probable major parent with 98.4% similarity, and HCoV-NL63-Netherlands (Accession# NC_005831) was identified as a minor parent. HCoV-NL63-Japan (Accession# LC488388) was identified as a recombinant in GENCONV event number 1 (Figure 6b). The recombination breakpoints start at nucleotide position 142 in the alignment with a 95% confidence interval (CI) and end at the 1082 nucleotide position with a 99% CI and with a Bonferroni-corrected *p*-value of 0.05 (Figure 6c).

## 4. Discussion

HCoVs are serious pathogens associated with human and animal diseases, causing respiratory illnesses globally [3]. The monitoring of HCoV infections at a molecular level with an emphasis on the genome and phylogeny enables us to elucidate the emergence of new variants/strains that may infect and cause diseases to new hosts, including animals and humans at different geographic locations. Seasonal HCoVs such as HCoV-NL63, -229E, -OC43, and -HKU1 cause only seasonal infections, while SARS-CoV-1, MERS-CoV, and SARS-CoV-2 are known to cause respiratory illnesses throughout the year [2,22,47,48]. Genetic changes in the viral genomes lead to disruption of the virus-and-host cell interactions, as well as changes in virus reproduction, virulence, pathogenicity, gene expression, and ultimately determine the outcome of the severe infection [47]. Due to the favorable climatic conditions, frequent recombination and mutation occurs in the coronavirus genome and new virus variants/strains and isolates emerge, which results in interspecies transmission and infection. Based on globally published reports, there are many hosts that have been identified as coronavirus reservoirs, such as bats, palm civets, raccoon dogs, and camels [49,50]. They use different receptors such as ACE-2, DPP4, and APN for the host cell attachment and entry to the host cell [51]. SARS-CoV-1 emerged in 2002–2003 and caused epidemics. This coronaviral genome had seven NT and six AA variations in its S gene that resulted in a low pathogenicity identified in palm civets and raccoon dogs. In 2003, a global pandemic was caused by a highly pathogenic SARS-CoV-1 disease. Based on sequence analyses, fourteen single-nucleotide variations caused changes into eleven AA changes, which led to its high pathogenicity. Another six nucleotide variations resulted in four amino acid variations and caused a global epidemic in 2003. Just after the first epidemic, in 2004, due to interspecies transmission and viral adaptation, four new cases of human infection were reported in China [52].

There are many published reports available describing the comparative study of genetic determinants with high- and low-virulence properties and mortality rates caused by HCoVs like SARS-CoV-1, SARS-CoV-2, and MERS-CoV in comparison to other HCoVs like HCoV-NL63, -229E, -OC43, and -HKU1 [53]. Global viromics studies of more than 3000 viral genomes collected from both humans and animals (SARS-CoV-1, MERS-CoV, and SARS-CoV-2) confirmed variations in four locations situated in the nucleoprotein gene (N) and S protein gene as compared to HCoV-NL63, -229E, -OC43, and -HKU1 [47,53]. The WHO has kept MERS-CoV on the priority list for performing detailed studies because of its continuous infection and spread to humans and camels in different locations [22]. Recently, the whole genome of MERS-CoV isolated from humans and camels was used for a global analysis of genetic diversity, and the results showed that three clades (A, B, and C) were generated in the software, and it was concluded that MERS-CoV and its new variants are circulating in camels [22]. Additionally, one more study from Ethiopia reported that the MERS-CoV infecting Ethiopian camels phylogenetically belongs to clade C2 and forms closed clusters with East African strains [36]. Due to their continuous exposure to infected animals, animal handlers can facilitate the transmission and introduction of moderately to highly virulent HCoVs in a specific geographic location. High mutation rates result in efficient virus transmission, severe infection, and easy host adaptation, and can cause global epidemics and pandemics. A similar phenomenon and favorable conditions were also observed in the city of Wuhan, China, which resulted in the emergence of SARS-CoV-2. Changes in the nucleotides and amino acids favor the emergence of new isolates, strains, mutations, or recombinant viruses, as has been observed and reported earlier in many published papers from Saudi Arabia for MERS-CoV and South Korean mutants, as well as other HCoVs such as SARS-CoV-1 and SARS-CoV-2 from different geographic locations [20,32,54,55,56,57,58]. The S protein gene mutations in other HCoVs have favored the high rate of interspecies transmission towards human receptors [59,60,61]. In the present study, the genome sequences of MERS-CoV showed less identity with SARS-CoV-1 but higher genome similarity with SARS-CoV-2. In the phylogenetic tree relationship analyses, it was observed that most of the virus isolates formed a closed cluster with their similar isolates like MERS-CoV, SARS-CoV-1, and SARS-CoV-2, as well as other HCoVs. Our data and findings are supported by earlier published reports about genetic diversity, phylogenetic analyses, and recombination analyses based on the MERS-CoV-S gene with selected HCoVs from both humans and camels [20,22]. Based on the phylogeny, it has been observed that even after the continuous import of infected camels from African countries, the MERS-CoV-African isolate did not establish itself in the KSA as both isolates formed separate clusters, and the Arabian MERS-CoV isolate is still circulating in camels [21,22,34,36].

Recombination is very crucial and play an important role in the emergence of a recombinant virus, new virus isolates, and variants/strains with novel properties during the life cycle of an HCoV with other co-circulating viruses in different hosts and locations. The published reports suggest that coronaviruses undergo rapid and frequent recombination, which leads to the emergence of new virus strains or variants with altered virulence and serious effects on cytokine storms [2]. The genomic alterations and gene flow of both humans and pathogens significantly favor the spread of pathogenic organisms from one to another location, as well as interspecies and intraspecies transmission [62,63]. It has been observed that the rate of mutations in CoVs, including HCoVs, is high in comparison to other ssRNA and DNA viruses [64,65]. In 2015, an outbreak of MERS-CoV was reported due to the emergence of a recombinant virus isolate with the co-circulation of HCoVs and MERS-CoV in Saudi Arabia. The co-circulation of HCoVs favors genomic recombination with MERS-CoV, which infects both humans and camels, resulting in the emergence of a novel recombinant virus that was lethal to humans [4,22,34,58].

The recombination patterns, breakpoints, and events provide very useful information during viral outbreaks caused by one or more viruses or any other recombinant as well as variant viruses. The identification of recombination events leads to identifying new variants or recombinants that have other properties, such as altered transmission patterns, virus replication, and infectivity, as well as epidemiological fitness to sustain the virus isolates or variants in different environmental conditions. Recombination events may take place during the evolution and transmission of HCoVs. There are various published reports (in silico and in vivo) about recombination events in SARS-CoV-2 [66,67]. Recently, in one study, a Recombination Inference using Phylogenetic Patterns (RIPPLES) program was developed to detect recombination events in large mutation-annotated tree (MAT) files. This program breaks the sequences into two distinct fragments based on the recombination and mutation in the sequence, and two breakpoints are separated. By using this program, a total of six hundred and six events of recombination were identified, and it was concluded that SARS-CoV-2 genomes exhibit recombination in the S gene [68]. There are several reports about the recombination between Alpha and Delta, Beta and Delta, and Omicron BA.1 and BA.1–BA.2 recombinants [41]. Additionally, in another report published by Preska Steinberg in 2023, the ORF1ab and S genes showed a high frequency of recombination when analyzed in 191 SARS-CoV-2 and related genomes [69]. The positive selection site of the MERS-CoV-S gene in camels has been found to favor host jumping and human infection. As it has been observed in MERS-CoV, the frequent recombination breakpoint occurs in the ORF1b/S gene, while in SARS-CoV-2, the S gene shows major variations and recombination, which has led to the emergence of new variants globally [22,34]. The recombination pattern of MERS-CoV with other selected HCoVs indicates co-infections with different MERS-CoV variants in camels, while the SARS-CoV-2 recombination pattern indicates that HCoV-NL63-USA is the probable major parent and HCoV-NL63-Netherlands is the minor parent. HCoV-NL63-Japan was identified as a recombinant in GENCONV event number 1 [21,22,34].

In this study, we have discussed the S protein gene based on genetic diversity, phylogenetic relationships, and recombination patterns and breakpoints, which will enable us to identify the emergence and spread of new recombinant viruses, variants, and isolates with an extended host range and novel properties. These variants or strains may infect multiple hosts in new geographical regions globally. These findings suggest that more elaborate genetic analysis research is further required focusing on other geographical regions as well as the full genome of each HCoV. In the future, for the detailed study of HCoVs, genomic analyses are required to understand the emergence and spread of new variants/isolates/strains of HCoV in dromedaries, pangolins, bats, humans, and other unidentified alternative hosts. There are still many knowledge gaps left requiring detailed information on both MERS-CoV and SARS-CoV-2, and comprehensive genotypic studies and follow-up analyses are needed, which will provide a clue as to whether asymptomatic MERS-CoV and SARS-CoV-2 infections in camels and humans, as well as in other hosts, are currently developing locally and globally. The generated data will aid in understanding how genetic diversity, selection, and recombination play important roles in modifying and molding the genetic structure of a specific virus that may lead to the emergence of new pandemics or epidemics.

## 5. Conclusions

This study was based on the genetic diversity and recombination pattern analysis of the MERS-CoV and SARS-CoV-2-S protein genes compared with other HCoVs. Based on the data generated, it is concluded that these S protein genes have significant diversity and a different recombination pattern when compared to other HCoVs. Continuous monitoring and detailed genetic diversity and phylogenetic analysis studies are required to understand this virus’s evolution, interspecies transmission, host adaptation, virulence, disease severity, and genetic relationship with other HCoVs. This study will be highly helpful for combatting any further outbreaks. Our analyses of recombination patterns will be beneficial for understanding the emergence of any new recombinant viruses or variants. These genetic diversity analyses and the identification of any mutation in the viral genome can be used to design and develop an effective vaccine for broad-spectrum protection against HCoVs globally.

## Figures and Tables

**Figure 1 biology-13-00282-f001:**
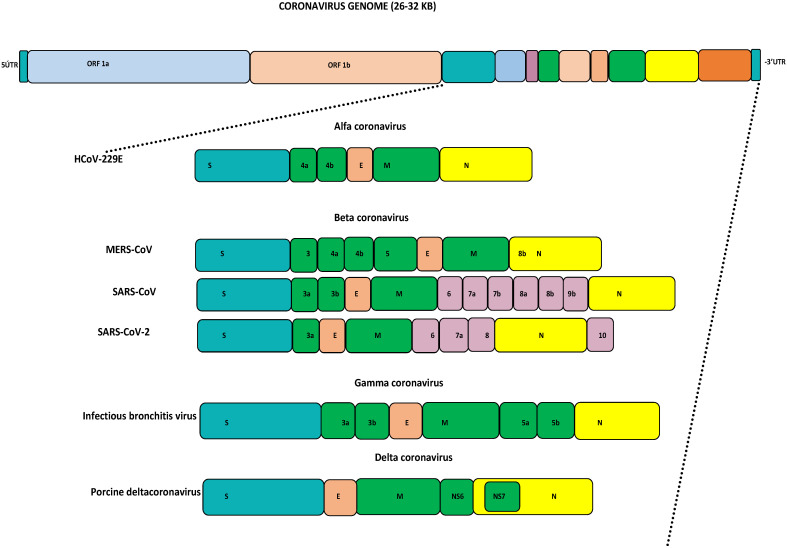
Genome organization of coronaviruses: **HCoV**: human coronavirus; **MERS-CoV**: Middle East respiratory syndrome coronavirus; **SARS-CoV**: severe acute respiratory syndrome coronavirus; **SARS-CoV-2**: severe acute respiratory syndrome coronavirus 2.

**Figure 2 biology-13-00282-f002:**
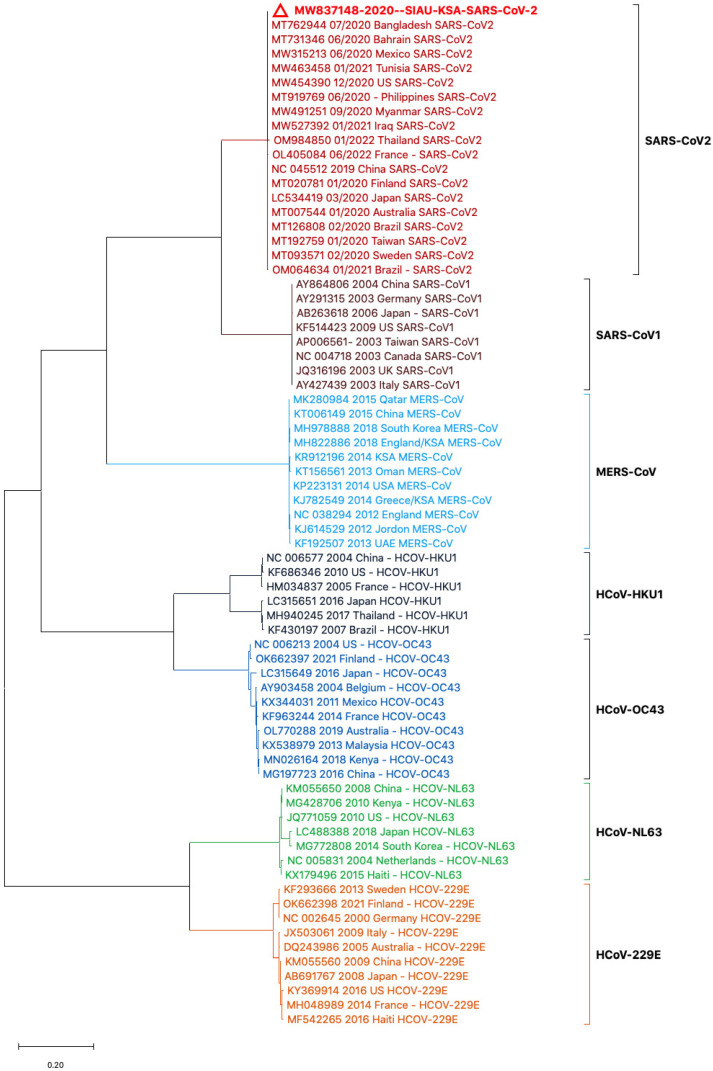
Phylogeny based on the nucleotide (NT) sequences of the S protein gene of selected HCoVs. The red triangle denotes the SARS-CoV-2 genome sequences from SIAU-KSA.

**Figure 3 biology-13-00282-f003:**
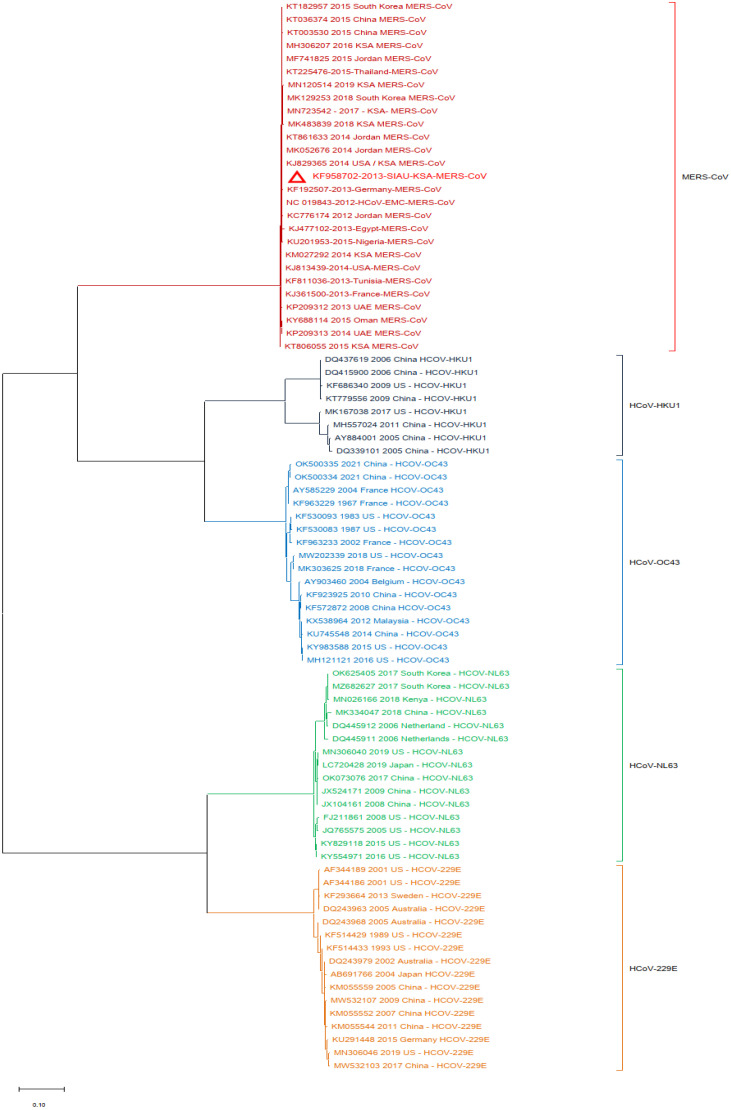
Phylogeny constructed by using the nucleotide (NT) sequences of the MERS-CoV-S protein gene with selected HCoVs and without SARS-CoV-1 and SARS-CoV-2. The red triangle denotes the MERS-CoV genome sequences from SIAU-KSA.

**Figure 4 biology-13-00282-f004:**
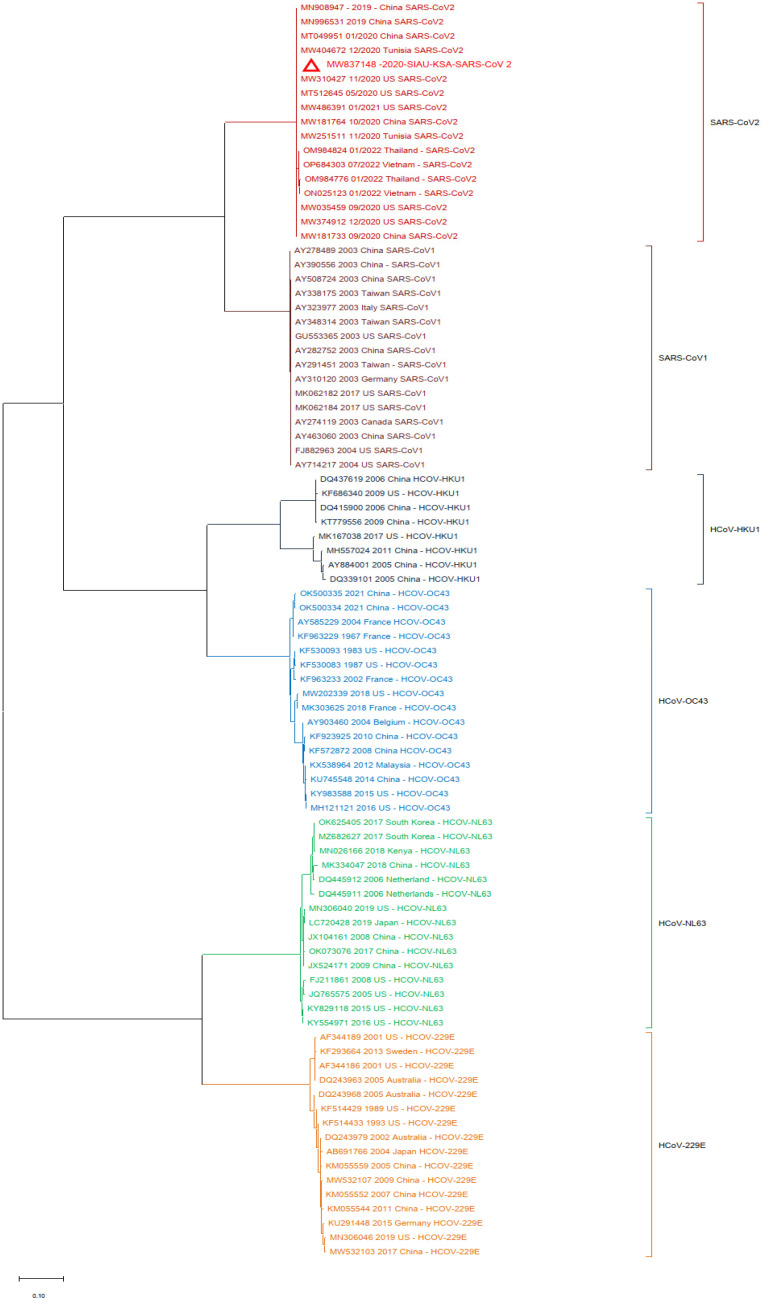
Phylogeny according to the nucleotide (NT) sequences of the SARS-CoV-2-S protein gene with selected HCoVs without MERS-CoV. The red triangle denotes the SARS-CoV-2 genome sequences from SIAU-KSA.

**Figure 5 biology-13-00282-f005:**
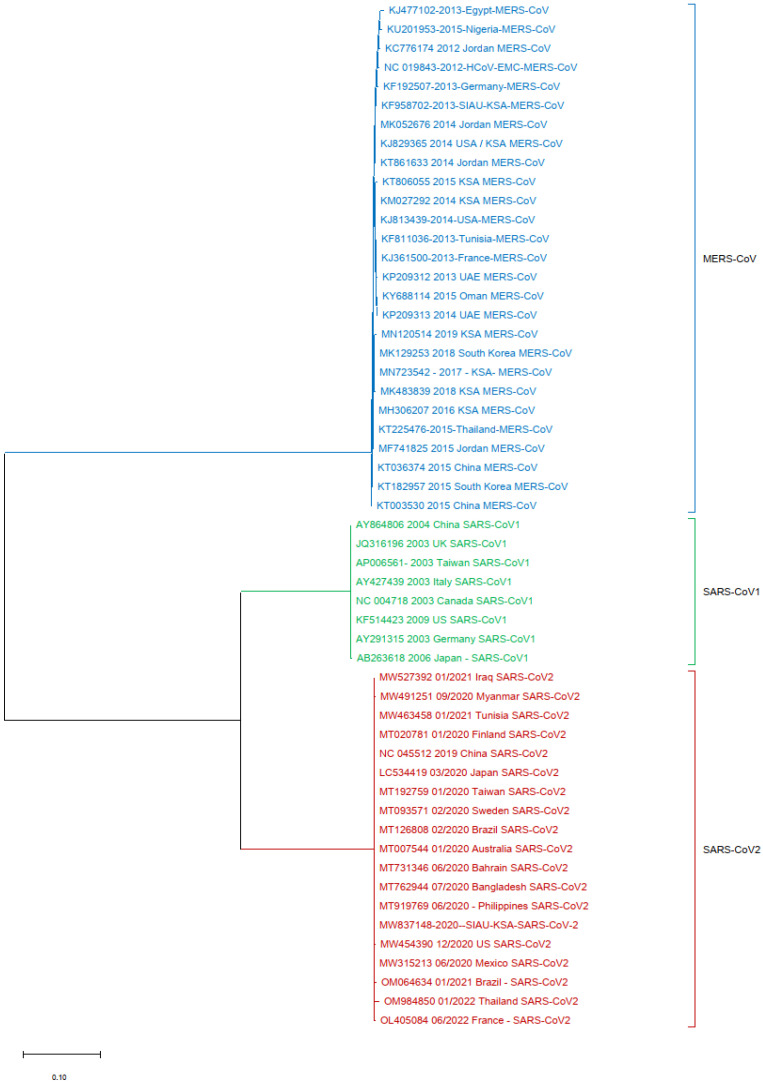
Phylogeny tree based only on the nucleotide (NT) sequences of the S protein gene of MERS-CoV with SARS-CoV-1 and SARS-CoV-2.

**Figure 6 biology-13-00282-f006:**
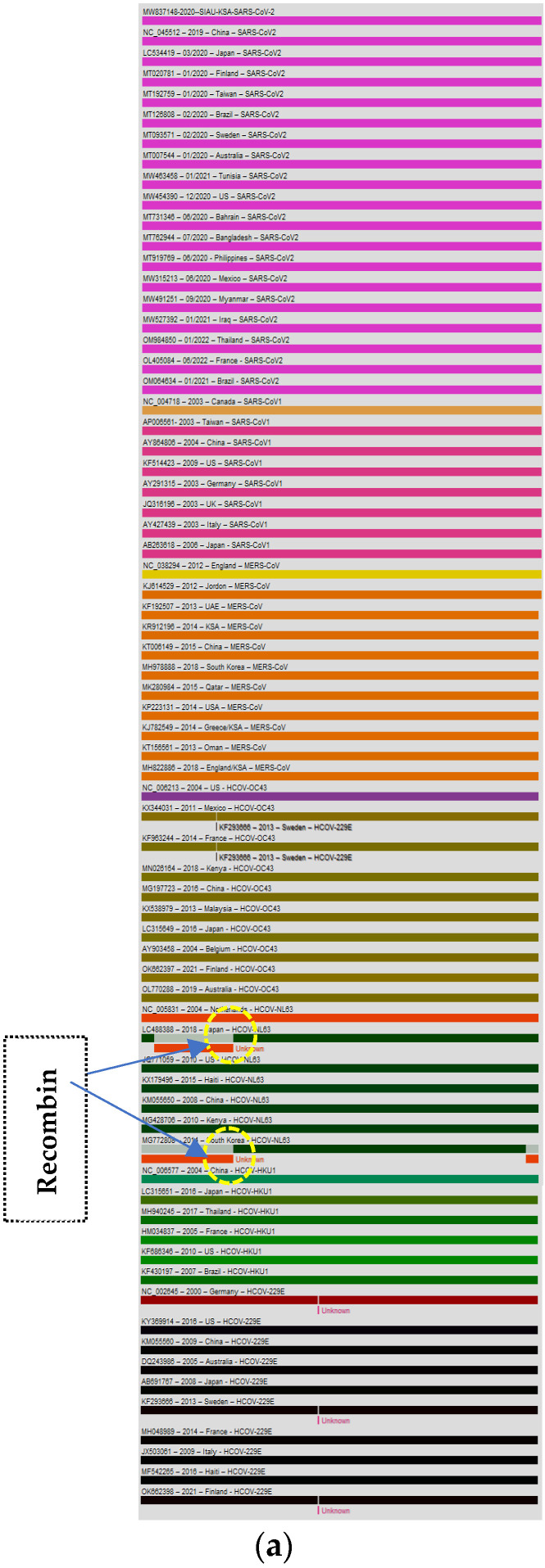
(**a**). Recombination pattern and breakpoints of the SARS-CoV-2-S protein gene (MW837148) with selected HCoVs. (**b**). Recombination events of the SARS-CoV-2-S protein gene (MW837148) with selected HCoVs. (**c**). Recombination breakpoints of the SARS-CoV-2-S protein gene (MW837148) with selected HCoVs.

**Table 1 biology-13-00282-t001:** Sequence identity matrix for the SARS-CoV-2-S protein gene (MW837148) with selected HCoVs.

S. No.	Accession Number	Virus	Country	Year	% Identity	
					NT	AA
1	NC_045512	SARS-CoV-2	China	2019	99.9	99.9
2	LC534419	SARS-CoV-2	Japan	2020	99.9	99.9
3	MT020781	SARS-CoV-2	Finland	2020	99.9	99.8
4	MT192759	SARS-CoV-2	Taiwan	2020	99.9	99.9
5	MT126808	SARS-CoV-2	Brazil	2020	99.9	99.9
6	MT093571	SARS-CoV-2	Sweden	2020	99.9	99.8
7	MT007544	SARS-CoV-2	Australia	2020	99.9	98.8
8	MW463458	SARS-CoV-2	Tunisia	2021	99.9	99.9
9	MW454390	SARS-CoV-2	USA	2020	99.9	99.8
10	MT731346	SARS-CoV-2	Bahrain	2020	99.9	99.9
11	MT762944	SARS-CoV-2	Bangladesh	2020	99.9	99.9
12	MT919769	SARS-CoV-2	Philippines	2020	99.9	99.9
13	MW315213	SARS-CoV-2	Mexico	2020	99.9	99.8
14	MW491251	SARS-CoV-2	Myanmar	2020	99.8	99.8
15	MW527392	SARS-CoV-2	Iraq	2021	99.9	99.8
16	OM984850	SARS-CoV-2	Thailand	2022	99.0	97.6
17	OL405084	SARS-CoV-2	France	2022	99.5	99.2
18	OM064634	SARS-CoV-2	Brazil	2021	99.8	99.5
19	NC_004718	SARS-CoV-1	Canada	2003	72.9	75.5
20	AP006561	SARS-CoV-1	Taiwan	2003	72.9	75.6
21	AY864806	SARS-CoV-1	China	2004	73.0	75.7
22	KF514423	SARS-CoV-1	USA	2009	72.9	75.6
23	AY291315	SARS-CoV-1	Germany	2003	73.0	75.5
24	JQ316196	SARS-CoV-1	UK	2003	72.9	75.6
25	AY427439	SARS-CoV-1	Italy	2003	72.9	75.6
26	AB263618	SARS-CoV-1	Japan	2006	72.9	75.3
27	NC_038294	MERS-CoV	England	2012	45.9	25.8
28	KJ614529	MERS-CoV	Jordon	2012	45.8	25.9
29	KF192507	MERS-CoV	UAE	2013	45.8	25.8
30	KR912196	MERS-CoV	KSA	2014	45.8	25.8
31	KT006149	MERS-CoV	China	2015	45.9	25.8
32	MH978888	MERS-CoV	South Korea	2018	45.8	25.8
33	MK280984	MERS-CoV	Qatar	2015	46.0	25.8
34	KP223131	MERS-CoV	USA	2014	45.9	25.9
35	KJ782549	MERS-CoV	Greece/KSA	2014	45.9	25.9
36	KT156561	MERS-CoV	Oman	2013	45.9	25.9
37	MH822886	MERS-CoV	England/KSA	2018	45.8	25.8
38	NC_006213	HCoV-OC43	USA	2004	40.9	26.3
39	KX344031	HCoV-OC43	Mexico	2011	40.8	26.4
40	KF963244	HCoV-OC43	France	2014	40.7	26.1
41	MN026164	HCoV-OC43	Kenya	2018	40.7	26.2
42	MG197723	HCoV-OC43	China	2016	40.8	26.2
43	KX538979	HCoV-OC43	Malaysia	2013	40.8	26.4
44	LC315649	HCoV-OC43	Japan	2016	40.7	26.2
45	AY903458	HCoV-OC43	Belgium	2004	40.5	26.3
46	OK662397	HCoV-OC43	Finland	2021	40.7	26.0
47	OL770288	HCoV-OC43	Australia	2019	40.7	26.4
48	NC_005831	HCoV-NL63	Netherlands	2004	37.5	19.8
49	LC488388	HCoV-NL63	Japan	2018	37.3	19.9
50	JQ771059	HCoV-NL63	USA	2010	37.7	19.6
51	KX179496	HCoV-NL63	Haiti	2015	37.5	19.6
52	KM055650	HCoV-NL63	China	2008	37.6	19.6
53	MG428706	HCoV-NL63	Kenya	2010	37.6	19.6
54	MG772808	HCoV-NL63	S. Korea	2014	37.5	20.1
55	NC_006577	HCoV-HKU1	China	2004	40.2	25.1
56	LC315651	HCoV-HKU1	Japan	2016	40.6	25.2
57	MH940245	HCoV-HKU1	Thailand	2017	40.7	25.2
58	HM034837	HCoV-HKU1	France	2005	40.1	25.0
59	KF686346	HCoV-HKU1	USA	2010	40.2	25.0
60	KF430197	HCoV-HKU1	Brazil	2007	40.7	25.1
61	NC_002645	HCoV-229E	Germany	2000	32.4	19.9
62	KY369914	HCoV-229E	USA	2016	32.3	19.8
63	KM055560	HCoV-229E	China	2009	32.5	19.8
64	DQ243986	HCoV-229E	Australia	2005	32.6	19.8
65	AB691767	HCoV-229E	Japan	2008	32.5	19.8
66	KF293666	HCoV-229E	Sweden	2013	32.4	19.8
67	MH048989	HCoV-229E	France	2014	32.5	19.8
68	JX503061	HCoV-229E	Italy	2009	32.6	19.9
69	MF542265	HCoV-229E	Haiti	2016	32.4	19.8
70	OK662398	HCoV-229E	Finland	2021	32.4	19.9

**Table 2 biology-13-00282-t002:** Sequence identity matrix for the MERS-CoV-S protein gene (NC_019843) with selected HCoVs excluding SARS-CoV 1 and SARS-CoV-2.

S. No.	Accession Number	Virus	Country	Year	% Identity	
					NT	AA
1	KF958702	MERS-CoV	KSA	2013	99.7	99.9
2	KM027292	MERS-CoV	KSA	2014	99.7	99.8
3	KT806055	MERS-CoV	KSA	2015	99.5	99.7
4	MH306207	MERS-CoV	KSA	2016	99.6	99.9
5	MN723542	MERS-CoV	KSA	2017	99.5	99.8
6	MK483839	MERS-CoV	KSA	2018	99.3	99.7
7	MN120514	MERS-CoV	KSA	2019	99.3	99.7
8	KJ829365	MERS-CoV	US/KSA	2014	99.7	99.7
9	KC776174	MERS-CoV	Jordan	2012	99.7	99.7
10	KT861633	MERS-CoV	Jordan	2014	99.7	99.8
11	MK052676	MERS-CoV	Jordan	2014	99.7	99.8
12	MF741825	MERS-CoV	Jordan	2015	99.6	99.9
13	KY688114	MERS-CoV	Oman	2015	99.6	99.7
14	KJ477102	MERS-CoV	Egypt	2013	99.2	99.0
15	KT182957	MERS-CoV	S. Korea	2015	99.5	99.7
16	MK129253	MERS-CoV	S. Korea	2018	99.5	99.9
17	KT036374	MERS-CoV	China	2015	99.5	99.9
18	KT003530	MERS-CoV	China	2015	99.4	99.7
19	KP209312	MERS-CoV	UAE	2013	99.6	99.7
20	KP209313	MERS-CoV	UAE	2014	99.6	99.7
21	KU201953	MERS-CoV	Nigeria	2015	99.5	99.4
22	KF811036	MERS-CoV	Tunisia	2013	99.7	99.7
23	KT225476	MERS-CoV	Thailand	2015	99.6	99.8
24	KJ361500	MERS-CoV	France	2013	99.7	99.7
25	KF192507	MERS-CoV	Germany	2013	99.7	99.8
26	KJ813439	MERS-CoV	USA	2014	99.7	99.8
27	KY983588	HCoV-OC43	USA	2015	46.1	28.8
28	MH121121	HCoV-OC43	USA	2016	46.1	28.8
29	KF530093	HCoV-OC43	USA	1983	46.2	28.7
30	MW202339	HCoV-OC43	USA	2018	45.7	28.5
31	KF530083	HCoV-OC43	USA	1987	46.1	28.7
32	KF923925	HCoV-OC43	China	2010	46.1	28.8
33	KF572872	HCoV-OC43	China	2008	46.0	28.9
34	KU745548	HCoV-OC43	China	2014	46.2	28.8
35	OK500335	HCoV-OC43	China	2021	45.8	28.5
36	OK500334	HCoV-OC43	China	2021	45.8	28.5
37	AY585229	HCoV-OC43	France	2004	46.0	28.6
38	KF963233	HCoV-OC43	France	2002	46.1	28.6
39	KF963229	HCoV-OC43	France	1967	46.0	28.6
40	MK303625	HCoV-OC43	France	2018	45.8	28.5
41	KX538964	HCoV-OC43	Malaysia	2012	46.1	28.8
42	AY903460	HCoV-OC43	Belgium	2004	45.8	28.9
43	FJ211861	HCoV-NL63	USA	2008	38.8	18.3
44	KY829118	HCoV-NL63	USA	2015	39.0	18.3
45	JQ765575	HCoV-NL63	USA	2005	38.9	18.3
46	KY554971	HCoV-NL63	USA	2016	39.0	18.3
47	MN306040	HCoV-NL63	USA	2019	39.0	18.3
48	MK334047	HCoV-NL63	China	2018	38.7	18.3
49	OK073076	HCoV-NL63	China	2017	39.0	18.4
50	JX524171	HCoV-NL63	China	2009	39.0	18.4
51	JX104161	HCoV-NL63	China	2008	39.0	18.4
52	DQ445911	HCoV-NL63	Netherlands	2006	38.7	18.3
53	DQ445912	HCoV-NL63	Netherland	2006	38.8	18.3
54	LC720428	HCoV-NL63	Japan	2019	39.0	18.3
55	OK625405	HCoV-NL63	S. Korea	2017	38.9	18.3
56	MZ682627	HCoV-NL63	S. Korea	2017	38.9	18.3
57	MN026166	HCoV-NL63	Kenya	2018	38.8	18.3
58	DQ437619	HCoV-HKU1	China	2006	45.7	28.1
59	KT779556	HCoV-HKU1	China	2009	45.6	27.9
60	AY884001	HCoV-HKU1	China	2005	45.5	28.0
61	DQ339101	HCoV-HKU1	China	2005	45.5	28.1
62	DQ415900	HCoV-HKU1	China	2006	45.7	2.8.1
63	MH557024	HCoV-HKU1	China	2011	45.3	27.9
64	MK167038	HCoV-HKU1	USA	2017	45.5	28.0
65	KF686340	HCoV-HKU1	USA	2009	45.7	28.1
66	KF514433	HCoV-229E	USA	1993	34.7	18.2
67	MN306046	HCoV-229E	USA	2019	34.7	18.2
68	AF344189	HCoV-229E	USA	2001	34.7	18.1
69	AF344186	HCoV-229E	USA	2001	34.7	18.2
70	KF514429	HCoV-229E	USA	1989	34.7	18.2
71	KM055559	HCoV-229E	China	2005	34.9	18.3
72	MW532107	HCoV-229E	China	2009	34.7	18.2
73	MW532103	HCoV-229E	China	2017	34.7	18.1
74	KM055544	HCoV-229E	China	2011	34.8	18.2
75	KM055552	HCoV-229E	China	2007	34.8	18.2
76	DQ243963	HCoV-229E	Australia	2005	34.7	18.2
77	DQ243979	HCoV-229E	Australia	2002	34.8	18.3
78	DQ243968	HCoV-229E	Australia	2005	34.6	18.2
79	KU291448	HCoV-229E	Germany	2015	34.8	18.1
80	AB691766	HCoV-229E	Japan	2004	34.7	18.2
81	KF293664	HCoV-229E	Sweden	2013	34.7	18.3

**Table 3 biology-13-00282-t003:** Sequence identity matrix for the SARS-CoV-2-S protein gene (MW837148) with selected HCoVs excluding MERS-CoV.

S. No.	Accession Number	Virus	Country	Year	% Identity	
					NT	AA
1	MW181733	SARS-CoV-2	China	2020	99.9	99.9
2	MW181764	SARS-CoV-2	China	2020	99.9	99.9
3	MN908947	SARS-CoV-2	China	2019	99.9	99.9
4	MT049951	SARS-CoV-2	China	2020	99.9	99.8
5	MN996531	SARS-CoV-2	China	2019	99.9	99.9
6	MT512645	SARS-CoV-2	USA	2020	99.9	99.9
7	MW035459	SARS-CoV-2	USA	2020	99.9	99.9
8	MW310427	SARS-CoV-2	USA	2020	99.9	99.9
9	MW374912	SARS-CoV-2	USA	2020	99.9	99.8
10	MW486391	SARS-CoV-2	USA	2021	99.9	99.9
11	MW251511	SARS-CoV-2	Tunisia	2020	99.9	99.8
12	MW404672	SARS-CoV-2	Tunisia	2020	99.9	99.8
13	OM984824	SARS-CoV-2	Thailand	2022	99.0	97.6
14	OM984776	SARS-CoV-2	Thailand	2022	98.5	96.9
15	OP684303	SARS-CoV-2	Vietnam	2022	98.8	97.4
16	ON025123	SARS-CoV-2	Vietnam	2022	98.6	96.8
17	AY291451	SARS-CoV1	Taiwan	2003	72.9	75.6
18	AY274119	SARS-CoV1	Canada	2003	72.9	75.5
19	AY463060	SARS-CoV1	China	2003	72.9	75.4
20	AY278489	SARS-CoV1	China	2003	72.9	75.7
21	AY390556	SARS-CoV1	China	2003	73.0	75.8
22	AY508724	SARS-CoV1	China	2003	72.9	75.5
23	AY282752	SARS-CoV1	China	2003	72.9	75.6
24	FJ882963	SARS-CoV1	USA	2004	72.9	75.6
25	GU553365	SARS-CoV1	USA	2003	72.9	75.6
26	AY714217	SARS-CoV1	USA	2004	72.9	75.5
27	MK062182	SARS-CoV1	USA	2017	72.9	75.5
28	MK062184	SARS-CoV1	USA	2017	72.9	75.5
29	AY348314	SARS-CoV1	Taiwan	2003	72.9	75.6
30	AY338175	SARS-CoV1	Taiwan	2003	72.9	75.6
31	AY310120	SARS-CoV1	Germany	2003	72.9	75.5
32	AY323977	SARS-CoV1	Italy	2003	72.9	75.6
33	KY983588	HCoV-OC43	USA	2015	42.4	26.4
34	MH121121	HCoV-OC43	USA	2016	42.4	26.4
35	KF530093	HCoV-OC43	USA	1983	42.7	26.6
36	MW202339	HCoV-OC43	USA	2018	42.7	26.5
37	KF530083	HCoV-OC43	USA	1987	42.7	26.6
38	KF923925	HCoV-OC43	China	2010	42.4	26.5
39	KF572872	HCoV-OC43	China	2008	42.5	26.5
40	KU745548	HCoV-OC43	China	2014	42.4	26.4
41	OK500335	HCoV-OC43	China	2021	42.6	26.0
42	OK500334	HCoV-OC43	China	2021	42.6	26.0
43	AY585229	HCoV-OC43	France	2004	42.7	26.5
44	KF963233	HCoV-OC43	France	2002	42.8	26.5
45	KF963229	HCoV-OC43	France	1967	42.7	26.5
46	MK303625	HCoV-OC43	France	2018	42.8	26.5
47	KX538964	HCoV-OC43	Malaysia	2012	42.4	26.4
48	AY903460	HCoV-NL63	Belgium	2004	42.4	26.4
49	FJ211861	HCoV-NL63	USA	2008	38.2	18.7
50	KY829118	HCoV-NL63	USA	2015	38.5	18.6
51	JQ765575	HCoV-NL63	USA	2005	38.3	18.6
52	KY554971	HCoV-NL63	USA	2016	38.6	18.6
53	MN306040	HCoV-NL63	USA	2019	38.1	18.5
54	MK334047	HCoV-NL63	China	2018	38.2	18.5
55	OK073076	HCoV-NL63	China	2017	38.3	18.6
56	JX524171	HCoV-NL63	China	2009	38.2	18.6
57	JX104161	HCoV-NL63	China	2008	38.3	18.6
58	DQ445911	HCoV-NL63	Netherlands	2006	38.4	18.7
59	DQ445912	HCoV-NL63	Netherland	2006	38.5	18.7
60	LC720428	HCoV-NL63	Japan	2019	38.2	18.6
61	OK625405	HCOV-NL63	S. Korea	2017	38.3	18.7
62	MZ682627	HCoV-NL63	S. Korea	2017	38.3	18.7
63	MN026166	HCoV-NL63	Kenya	2018	38.4	18.7
64	DQ437619	HCoV-HKU1	China	2006	42.2	24.8
65	KT779556	HCoV-HKU1	China	2009	42.2	24.8
66	AY884001	HCoV-HKU1	China	2005	42.2	25.0
67	DQ339101	HCoV-HKU1	China	2005	42.3	25.1
68	DQ415900	HCoV-HKU1	China	2006	42.2	24.8
69	MH557024	HCoV-HKU1	China	2011	42.2	25.0
70	MK167038	HCoV-HKU1	USA	2017	42.0	25.0
71	KF686340	HCoV-HKU1	USA	2009	42.2	24.8
72	KF514433	HCoV-229E	USA	1993	33.0	19.6
73	MN306046	HCoV-229E	USA	2019	32.7	19.6
74	AF344189	HCoV-229E	USA	2001	32.8	19.6
75	AF344186	HCoV-229E	USA	2001	32.9	19.5
76	KF514429	HCoV-229E	USA	1989	33.0	19.6
77	KM055559	HCoV-229E	China	2005	32.9	19.6
78	MW532107	HCoV-229E	China	2009	32.8	19.6
79	MW532103	HCoV-229E	China	2017	32.8	19.6
80	KM055544	HCoV-229E	China	2011	32.9	19.6
81	KM055552	HCoV-229E	China	2007	32.9	19.6
82	DQ243963	HCoV-229E	Australia	2005	32.9	19.5
83	DQ243979	HCoV-229E	Australia	2002	32.9	19.6
84	DQ243968	HCoV-229E	Australia	2005	32.9	19.6
85	KU291448	HCoV-229E	Germany	2015	32.8	19.5
86	AB691766	HCoV-229E	Japan	2004	32.8	19.6
87	KF293664	HCoV-229E	Sweden	2013	32.8	19.4

**Table 4 biology-13-00282-t004:** Sequence identity matrix for the MERS-CoV-S protein gene (NC_019843) with selected SARS-CoV 1 and SARS-CoV-2.

S. No.	Accession Number	Virus	Country	Year	% Identity	
					NT	AA
1	KF958702	MERS-CoV	KSA	2013	99.7	99.9
2	KM027292	MERS-CoV	KSA	2014	99.7	99.8
3	KT806055	MERS-CoV	KSA	2015	99.5	99.7
4	MH306207	MERS-CoV	KSA	2016	99.6	99.9
5	MN723542	MERS-CoV	KSA	2017	99.5	99.8
6	MK483839	MERS-CoV	KSA	2018	99.3	99.7
7	MN120514	MERS-CoV	KSA	2019	99.3	99.7
8	KJ829365	MERS-CoV	USA/KSA	2014	99.7	99.7
9	KC776174	MERS-CoV	Jordan	2012	99.7	99.7
10	KT861633	MERS-CoV	Jordan	2014	99.7	99.8
11	MK052676	MERS-CoV	Jordan	2014	99.7	99.8
12	MF741825	MERS-CoV	Jordan	2015	99.6	99.9
13	KY688114	MERS-CoV	Oman	2015	99.6	99.7
14	KJ477102	MERS-CoV	Egypt	2013	99.2	99.0
15	KT182957	MERS-CoV	S. Korea	2015	99.5	99.7
16	MK129253	MERS-CoV	S. Korea	2018	99.5	99.9
17	KT036374	MERS-CoV	China	2015	99.5	99.9
18	KT003530	MERS-CoV	China	2015	99.4	99.7
19	KP209312	MERS-CoV	UAE	2013	99.6	99.7
20	KP209313	MERS-CoV	UAE	2014	99.6	99.7
21	KU201953	MERS-CoV	Nigeria	2015	99.5	99.4
22	KF811036	MERS-CoV	Tunisia	2013	99.7	99.7
23	KT225476	MERS-CoV	Thailand	2015	99.6	99.8
24	KJ361500	MERS-CoV	France	2013	99.7	99.7
25	KF192507	MERS-CoV	Germany	2013	99.7	99.8
26	KJ813439	MERS-CoV	USA	2014	99.7	99.8
27	MW837148	SARS-CoV-2	KSA	2020	44.2	26.7
28	NC_045512	SARS-CoV-2	China	2019	44.2	26.7
29	LC534419	SARS-CoV-2	Japan	2020	44.2	26.7
30	MT020781	SARS-CoV-2	Finland	2020	44.2	26.7
31	MT192759	SARS-CoV-2	Taiwan	2020	44.2	26.7
32	MT126808	SARS-CoV-2	Brazil	2020	44.2	26.7
33	MT093571	SARS-CoV-2	Sweden	2020	44.2	26.6
34	MT007544	SARS-CoV-2	Australia	2020	44.2	26.7
35	MW463458	SARS-CoV-2	Tunisia	2021	44.2	26.7
36	MW454390	SARS-CoV-2	USA	2020	44.2	26.7
37	MT731346	SARS-CoV-2	Bahrain	2020	44.2	26.7
38	MT762944	SARS-CoV-2	Bangladesh	2020	44.2	26.7
39	MT919769	SARS-CoV-2	Philippines	2020	44.2	26.7
40	MW315213	SARS-CoV-2	Mexico	2020	44.2	26.7
41	MW491251	SARS-CoV-2	Myanmar	2020	44.2	26.7
42	MW527392	SARS-CoV-2	Iraq	2021	44.2	26.7
43	OM984850	SARS-CoV-2	Thailand	2022	44.0	26.6
44	OL405084	SARS-CoV-2	France	2022	44.0	26.6
45	OM064634	SARS-CoV-2	Brazil	2021	44.1	26.6
46	NC_004718	SARS-CoV1	Canada	2003	44.8	26.5
47	AP006561	SARS-CoV1	Taiwan	2003	44.9	26.5
48	AY864806	SARS-CoV1	China	2004	44.8	26.4
49	KF514423	SARS-CoV1	USA	2009	44.8	26.5
50	AY291315	SARS-CoV1	Germany	2003	44.9	26.6
51	JQ316196	SARS-CoV1	UK	2003	44.9	26.5
52	AY427439	SARS-CoV1	Italy	2003	44.9	26.5
53	AB263618	SARS-CoV1	Japan	2006	44.9	26.5

## Data Availability

The data presented in this study are available on request from the corresponding author.

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
