# Peer review of "Genomic Diversity and Recombination Analysis of the Spike Protein Gene from Selected Human Coronaviruses"

_biology, 2024, doi:10.3390/biology13040282_

Round 1

Reviewer 1 Report

Comments and Suggestions for Authors

The significance of this problem is very important globally. Assessment of genetic diversity of the MERS-CoV, SARS-CoV-2-S genes and other HCoVs is required to understand the virus's evolution, relationship between acquisition of new mutations and characteristics giving viruses advantage in evolution.

The title is informative, the aim is clear. The introduction of this article successfully explains why the understanding these mechanisms that underlie genetic diversity and the emergence of new mutations in viruses is very important.

Here are some suggestions that would improve the manuscript:

Chapter “Introduction”:

1.According to the context in the Introduction, the authors can give a scheme/figure of the the genomes of some important alpha-CoVs, beta-Covs (SARS-CoV-1, MERS-CoV and SARS-CoV-2), delta-Covs and gamma-CoVs. The 3′ terminus encodes structural proteins: envelope glycoprotein (S), envelope (E), membrane (M), and nucleocapsid (N). In addition to them, there are accessory genes that are species-specific but not dispensable for virus replication.

2. line 79 - "the highest mutation " - paraphrase )"the highest mutation rate site.., or "site where the mutations arise with high frequency",..

Chapter “Materials and Methods”

In the methodology section - the methods adequately described.

Мaybe somewhere you can describe more precisely how many and how the samples are divided, from which areas, how they are selected and if there are more from some geographic areas compared to others..

The “Results” -  clearly presented and illustrated with tables.

The “Discussion - The discussion volume is quite large compared to the other chapters. The first two paragraphs repeat some findings about the origin and spread of coronaviruses that are in the introduction. Precise this part of the discussion.

line 295 - "The genetic changes in the viral genomes lead to disruption of virus and host cell interactions.......add ....CHANGES "in virus reproduction, virulence, pathogenicity, gene expression, and ultimately determine the outcome of the infection".

The “Conclusion section, on the other hand, can be improved

To highlight and illustrate the the importance of the problem, it is good to explain how understanding of these recombinant mechanisms, the basis of genetic diversity and the appearance of new mutations in viruses, can help prevent future pandemics.

Comments on the Quality of English Language

minor editiing required

Author Response

Reviewer-1

Comments and Suggestions for Authors

The significance of this problem is very important globally. Assessment of genetic diversity of the MERS-CoV, SARS-CoV-2-S genes and other HCoVs is required to understand the virus's evolution, relationship between acquisition of new mutations and characteristics giving viruses advantage in evolution.

Response: Thanks for your understanding and good comments.

The title is informative, the aim is clear. The introduction of this article successfully explains why the understanding these mechanisms that underlie genetic diversity and the emergence of new mutations in viruses is very important.

Response: Thanks for your good comments and finding the novelty in this work

Here are some suggestions that would improve the manuscript:

Chapter “Introduction”:

1.According to the context in the Introduction, the authors can give a scheme/figure of the the genomes of some important alpha-CoVs, beta-Covs (SARS-CoV-1, MERS-CoV and SARS-CoV-2), delta-Covs and gamma-CoVs. The 3′ terminus encodes structural proteins: envelope glycoprotein (S), envelope (E), membrane (M), and nucleocapsid (N). In addition to them, there are accessory genes that are species-specific but not dispensable for virus replication.

Response: As per your comments and suggestions; we have made a figure and presented in the revised MS.

  1. line 79 - "the highest mutation " - paraphrase )"the highest mutation rate site.., or "site where the mutations arise with high frequency",..

Response: Thanks for your critical observations and comments; we have edited as you suggested. 

Chapter “Materials and Methods”

In the methodology section - the methods adequately described.

Мaybe somewhere you can describe more precisely how many and how the samples are divided, from which areas, how they are selected and if there are more from some geographic areas compared to others..

Response:  Thanks for your suggestions; We have adequately described the information as suggested.

 The “Results” -  clearly presented and illustrated with tables.

Response:  Thanks for your good comments

The “Discussion” - The discussion volume is quite large compared to the other chapters. The first two paragraphs repeat some findings about the origin and spread of coronaviruses that are in the introduction. Precise this part of the discussion.

Response:  Thanks for your suggestions; we have removed the duplicate words and reduced the information in this section.

line 295 - "The genetic changes in the viral genomes lead to disruption of virus and host cell interactions.......add ....CHANGES "in virus reproduction, virulence, pathogenicity, gene expression, and ultimately determine the outcome of the infection".

Response:  Thanks for your suggestions; We have added the text.

The “Conclusion” section, on the other hand, can be improved.

To highlight and illustrate the the importance of the problem, it is good to explain how understanding of these recombinant mechanisms, the basis of genetic diversity and the appearance of new mutations in viruses, can help prevent future pandemics.

Response:  Thanks for your suggestions; We have improved the conclusion section as suggested.

Reviewer 2 Report

Comments and Suggestions for Authors

The authors we have evaluated the S gene-based genetic diversity, phylogenetic relationship, and recombination patterns of selected HCoVs with emphasis on MERS-CoV and SARS-CoV-2 to elucidate the possible emergence of new variants/strains of coronavirus. The main findings included that MERS-CoV and SARS-CoV-2 have significant sequence identity with the selected HCoVs, while a separate cluster for each HCoV is revealed on the phylogenetic tree. In particular, the HCoV-NL63-Japan was identified as a probable recombinant, with the HCoV-NL63-USA and HCoV-NL63-Netherlands identified as a major and a minor parent, respectively. The authors claimed that the possible emergence of new strains of HCoVs is imminent.

This is an important topic to pursue following the recent covid worldwide. The findings are of significant importance for the investigation of these pathogenic viruses. The overall presentation is fine, though I do see some grammatical and editorial errors throughout the manuscript.

I have no major technical concerns but some comments and suggestions that I would like to ask for authors to clarify in case a revision is requested by the editor.

Title:

No need to provide abbreviation in the title.

Introduction:

I believe that the authors have provided sufficient background.

Line 68, a reference is needed for the first sentence.

Lines 95 and 97, Li et al. and Zhu et al. both need references.

Line 105: here, the authors should explicitly describe the goals of this study, instead of simply saying what they did.

Materials and Methods:

The authors need to supply more detailed explanations of the methodologies used in this study, e.g., the phylogenetic analysis was not indicated at all in this section but the conclusions of this study are largely based on the phylogenetic analysis.

Results:

I believe that the results are presented using appropriate tables and figures. The results are concisely described.

The red triangle symbol in figures 1-3 needs to be explained in the figure legends.

Discussion:

I believe that the authors did a good job evaluating their results in relation to the similar studies in the literature. I enjoyed reading this section. The logic flow reads well and the comprehensive discussion of their results make their conclusions based on available data strongly supported.

Comments on the Quality of English Language

The English is fine, though some grammatical and editorial errors are detected...

Author Response

Reviewer-2

Comments and Suggestions for Authors

The authors we have evaluated the S gene-based genetic diversity, phylogenetic relationship, and recombination patterns of selected HCoVs with emphasis on MERS-CoV and SARS-CoV-2 to elucidate the possible emergence of new variants/strains of coronavirus. The main findings included that MERS-CoV and SARS-CoV-2 have significant sequence identity with the selected HCoVs, while a separate cluster for each HCoV is revealed on the phylogenetic tree. In particular, the HCoV-NL63-Japan was identified as a probable recombinant, with the HCoV-NL63-USA and HCoV-NL63-Netherlands identified as a major and a minor parent, respectively. The authors claimed that the possible emergence of new strains of HCoVs is imminent.

 This is an important topic to pursue following the recent covid worldwide. The findings are significant for the investigation of these pathogenic viruses. The overall presentation is fine, though I do see some grammatical and editorial errors throughout the manuscript.

Response:  Thanks for your suggestions and finding some valuable information in our MS.We have rectified the grammatical errors in the  MS text.

I have no major technical concerns but some comments and suggestions that I would like to ask for authors to clarify in case a revision is requested by the editor.

 Title: No need to provide abbreviation in the title.

Response:  Thanks for your suggestions; We have removed the abbreviation from the title.

Introduction:

I believe that the authors have provided sufficient background.

 Response:  Thanks for your good comments.

Line 68, a reference is needed for the first sentence.

 Response:  Thanks for your suggestions; We have added the references as suggested

Lines 95 and 97, Li et al. and Zhu et al. both need references.

 Thanks for your suggestions; We have added the references as suggested.

Line 105: here, the authors should explicitly describe the goals of this study, instead of simply saying what they did.

Response:  Thanks for your suggestions; We have added the information about the objective of this study in this revised MS.

 Materials and Methods:

The authors need to supply more detailed explanations of the methodologies used in this study, e.g., the phylogenetic analysis was not indicated at all in this section, but the conclusions of this study are largely based on the phylogenetic analysis.

 Response:  Thanks for your suggestions; We have added the information about phylogenetic analysis in methodology section as suggested.

Results:

I believe that the results are presented using appropriate tables and figures. The results are concisely described.

 Response:  Thanks for your good commets.

The red triangle symbol in figures 1-3 needs to be explained in the figure legends.

 Response:  Thanks for your suggestions; We have explained the triangles as suggested.

Discussion:

I believe that the authors did a good job evaluating their results in relation to the similar studies in the literature. I enjoyed reading this section. The logic flow reads well, and the comprehensive discussion of their results make their conclusions based on available data strongly supported.

Response:  Thanks for your good comments and support

The English is fine, though some grammatical and editorial errors are detected...

Response:  Thanks for your suggestions; We have removed the grammatical errors in the revised text.